# Cartographer SLAM Method for Optimization with an Adaptive Multi-Distance Scan Scheduler

**Abdurahman Dwijotomo [1], Mohd Azizi Abdul Rahman [1,*], Mohd Hatta Mohammed Ariff [1], Hairi Zamzuri [2] and Wan Muhd Hafeez Wan Azree [2]**

1   Advanced Vehicle System Research Group, Universiti Teknologi Malaysia, 54100 Jalan Sultan Yahya Petra, Kuala Lumpur, Malaysia; adwijotomo2@live.utm.my (A.D.); mohdhatta.kl@utm.my (M.H.M.A.)
2   Emoovit Technology Sdn. Bhd., Level 1, Futurise Centre, Persiaran Apec 63000, Cyberjaya, Selangor, Malaysia; hairi@moovita.com (H.Z.); wanhafeez@moovita.com (W.M.H.W.A.)
*   Correspondence: azizi.kl@utm.my

**Abstract:** This paper presents the use of Google's simultaneous localization and mapping (SLAM) technique, namely Cartographer, and adaptive multistage distance scheduler (AMDS) to improve the processing speed. This approach optimizes the processing speed of SLAM which is known to have performance degradation as the map grows due to a larger scan matcher. In this proposed work, the adaptive method was successfully tested in an actual vehicle to map roads in real time. The AMDS performs a local pose correction by controlling the LiDAR sensor scan range and scan matcher search window with the help of scheduling algorithms. The scheduling algorithms manage the SLAM that swaps between short and long distances during map data collection. As a result, the algorithms efficiently improved performance speed similar to short distance LiDAR scans while maintaining the accuracy of the full distance of LiDAR. By swapping the scan distance of the sensor, and adaptively limiting the search size of the scan matcher to handle difference scan sizes, the pose's generation performance time is improved by approximately 16% as compared with a fixed scan distance, while maintaining similar accuracy.

**Keywords:** Cartographer; simultaneous localization and mapping (SLAM); LiDAR; scheduler

## 1. Introduction

Cartographer is a SLAM method developed by Google, which utilizes grid-based mapping together with a Ceres based scan matcher to reconstruct environment across various sensors configuration [1]. Initially, it was developed for portable applications to map unknown areas with a person using the inertial measurement unit (IMU) and LiDAR installed in a backpack. However, since the codes have been made public, Cartographer is now managed by an open source community, and various improvements have been made to widen the scope of applications including more substantial map support, more sensor integration, and other technological improvements which support intelligent robots. SLAM methods are not new research and are beyond the focus of this paper. However, this paper contributes to the improvisation of the current Cartographer or SLAM by using an adaptive multistage distance scheduler to reduce computational load while maintaining accuracy. The main contributions of this paper are as follows:

(a)   Optimizing Cartographer by using an adaptive multistage distance scheduler (AMDS) to increase time computation performance up to 20% while maintaining the pose generation and map accuracy similar to that of standard Cartographer;

(b)   Integrating Cartographer by using a car for online world mapping, which is crucial for the future autonomous vehicle navigation systems;

(c)    Investigating the performance of the proposed method by comparing it to the popular SLAM methods (LiDAR odometry and mapping (LOAM) and normal distribution transform (NDT) SLAM), in the actual world environment.

## 2. Related Works

In autonomous vehicle navigation systems, SLAM is a method used to generate map awareness of the vehicle. It is mainly used to know or locate the position of the vehicle body (i.e., similar to odometry) based on an environmental scan comparison with the generated map. Unlike odometry algorithms which use motion sensors (e.g., encoders, IMUs, etc.) to estimate pose, SLAM reduces or eliminates the drift issue seen in odometry. Drift error is an accumulation of small computational errors due to sensor accuracy specifications, electrical noises, temperature, etc., which when summed in odometry pose estimation can result in a linear increase of pose error based on travel distance. SLAM solves or reduces drift error of pose estimation by using an optimization technique using map data. This requires sensors which are able to sense the environment and object distances such as LiDAR or three-dimensional (3D) camera and sometimes additional sensors such as IMU encoders are needed [2,3]. The map is generated through the accumulation of scan data by this sensor during vehicle motion. Some of the SLAM methods also require odometry sensors such as IMU or wheel speed to provide better pose estimation accuracy. When the SLAM recognizes previously visited areas by comparing the environment with the map, it triggers pose correction to eliminate or reduce the drift error. Figure 1 illustrates the SLAM framework used in this work.

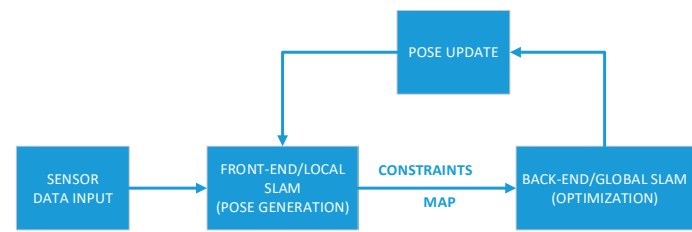

**Figure 1.** Simultaneous localization and mapping (SLAM) framework.

Although the SLAM method provides improved pose calculation for autonomous vehicles by recognizing previously recorded areas, there are still some issues that need to be addressed, for example, creating a real-time process in SLAM for mapping has some computation problems related to reduced performance due to the use of a scan matcher. A scan matcher is used to perform two different tasks. On the one hand, one task is the initial pose estimation, and on the other hand, a second task is the pose correction. The initial pose estimation compares the previous step of sensor scans of LiDAR or camera against current scan data. Thus, the disparity residual of current scan data versus the previous scan data in each step is used to estimate vehicle motion for the initial pose, similar to the odometry method. Over time, it eventually drifts due to the accumulation of error. The drift error is then corrected using the latter method, which is a scan matcher against the overall map. The process records the map by accumulating scan data in every step of vehicle motion. Using a larger scale scan matcher, unlike the former task, it compares the current scan with the whole map globally. When the scan matcher finds similarity above a certain threshold on the global map, the position of the scan and vehicle when drifting are aligned to a recognized area, and thus a pose correction scenario is created.

A small map runs fast enough with SLAM, however, overtime it will become cumbersome as the map grows larger during SLAM's map creation. As the map size becomes larger, it consumes more memory capacity and the scan matching performance becomes slower [4–6]. Some improvements have been introduced to improve computational performance. In [7–11], there are some SLAM methods that use specialized computer hardware for parallel programming, for example, a graphic processing unit (GPU). The GPU is much faster than the central processing unit (CPU) for processing a large dimension of matrix arrays or parallel executions. Map data in SLAM are similar to a multimedia image which

consists of a pixel in a multidimensional large array format. Meanwhile, the scan matcher also involves repetitive calculations to process the array data in various rotations and translations until similarity is found above a certain threshold. Thus, this kind of parallel computation is suitable to offload the workload through GPU. Although this type of computation is correct when offloaded to GPU, it should be noted that not all calculations are suitable and some must be offloaded to CPU when more serial processing power is needed. It also requires additional hardware which increases the cost of a GPU.

Some other techniques than GPU processing are done using a simplified scan or map dataset. Saving full map data in every scan sometimes is not efficient for a scan matcher. The higher map detail or resolution results in a higher dimension of total number of data to hold the map. Thus, it consumes more massive computational resources as compared with smaller maps. In [12–15], they introduced a simplified map by using a grid-based map. For every scan of the sensor, first, the data are down sampled in order to reduce map information. The scan is either down sampled to a two-dimensional (2D) grid space or to a 3D space using normal distribution transform (NDT) [16,17] clustering if the sensors are using 3D-based LiDAR. As the map becomes less complicated, the scan matcher processes the data faster, however, the accuracy of the low-resolution map largely depends on a map feature that stands out such as a structure, pole, etc., and the grid size of the map dictates the minimum distance error variance of the pose.

In addition to grid-based maps, there is also a feature- or topography-based map which saves some important features from the sensor scan rather than full-scan data such as planar surfaces, edge lines, landmark, etc. [18–21]. SLAM that uses a topography map requires feature pattern training to recognize the object from the scan and use it as a landmark on the map. While the vehicle is moving, SLAM records the position of landmarks or features and saves them inside the topography-based map. The pose is then calculated by associating the recognized object in each sensor scan against the related saved feature objects in the topography map. The pose generation is faster than that of a method utilizing a normal map as it skips plenty of information outside the referred features. However, if the environment lacks the data of the referred features, it can result in poor pose calculations.

There are other techniques to improve the performance of computational speed such as a multi-resolution map for branch and bound scan matching implemented in Google's Cartographer technique [1,22]. It is a scan matcher applied on a multi-resolution grid map. In Cartographer, map grids with lower resolution are utilized for the initial pose estimation. Later on, the initial pose accuracy is further increased using a scan matcher on higher resolution. Of course, by using known initial pose estimation, it can skip various pose iterations of the scan matcher on a higher resolution map. Thus, the computational speed is improved without sacrificing the accuracy of the map and pose generated.

In this paper, we propose another optimization method inspired by the multi-grid resolution map from Google's Cartographer technique. Instead of using multi-resolution maps, the proposed method uses a scheduler to filter distance size of the LiDAR sensor. It is known in SLAM that sensor distance can affect map-sampling size. A bigger map sampled helps reduce the drift error since more feature data are added, however it is not necessary to use full sensor data every time to create a bigger map. The scheduler works to bridge the sensor distance, which swaps between a small and full-scan size, and therefore the pose estimation using a scan matcher can be faster.

## 3. System Overview

The work presented in this study is based on a modified Google's Cartographer technique. Cartographer is a SLAM method which focuses on real-time performance and portability. It uses grid map-based representation with flexible resolution and sensor choices. The Cartographer technique is divided into two processes. The first process is called local SLAM which uses a Ceres scan matcher on a small map, known as a submap, to estimate the pose and orientation of the vehicle. The second process is known as global SLAM which optimizes the pose by utilizing a larger scan matcher on global maps (a global map generated from assembling all submaps). Some modifications are made to

the Cartographer's local SLAM integrated with the AMDS to control the sensor distances, and the scan matcher's search window during map generation and pose generation.

### 3.1. Multi-Distance Scheduler in LiDAR Scans

A multi-distance scheduler (MDS) is inspired from the idea of multiple resolution maps in Google's Cartographer utilizing a multi-stage scan matcher from low to high resolution. An MDS can be used to reduce computational cost as compared with using full high-resolution scans. It uses a scan matcher with low-resolution map data for the initial pose estimation, which then is used to bypass scan matcher iteration of every possible translation and rotation in a high-resolution map. Thus, it minimizes the performance penalty while maintaining accuracy.

The multi-distance approach is similar to the multi-resolution map. Instead of controlling resolution, we propose controlling sensor distance that generates point cloud frame ($\mathcal{F}$), whereby contains an array of object point ($\mathcal{F}$) detected from the LiDAR in local sensor coordinates. Limiting sensor distance results in different details of the scan, as shown in Figure 2.

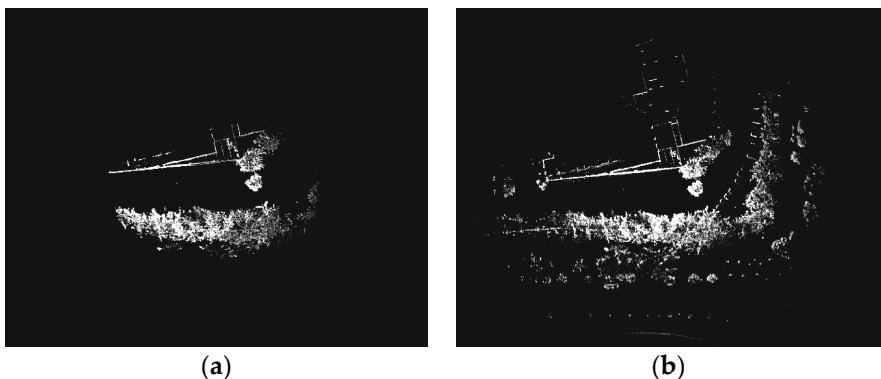

(**a**)                              (**b**)

**Figure 2.** Sensor scan state with different distances: (**a**) LiDAR scan 25 m distances and (**b**) LiDAR scan 60 m distances.

The controlling method is done via a scheduler, which swaps between two sensor states, namely a short distance state and a long distance state at a fixed frequency. It is known in scan matching algorithms that the larger the image data or map, the greater the amount of processing time required as compared with small images [23]. However, because the large images from the sensor scan capture more features, the computation results using these features generate less rotational and translational errors. Thus, it is hoped that the combination of these two scan states, i.e., short and long distance, achieves a similar result of increasing the processing speed performance while maintaining accuracy. Some additional modifications are needed to accommodate the proposed method. Table 1 shows the scheduler algorithm to control the distance of the scan. It uses a time variable which is based on the period for trigger. The period ratio is configured to, for example, 1:1, 1:2, 1:3, etc. The weight of scan matcher also needs to be weighted differently. A more extensive distance sensor scan requires higher weight on the SLAM pose estimation as compared with a shorter scan distance because a larger map contains more feature data, and thus is likely to have low covariance as more features are used to compute the pose.

**Table 1.** Psuedocode for a multi-distance scheduler.

**Function** Multi-distance scheduler (*PointsCloud_ℱ*, *Period_ratio_on*, *Max_Distances*, *Min_Distances)*
　　**if** (step scheduler == *Period_ratio_time_on*)
　　　　**for** all *p_point* in *PointsCloud_ℱ* **do**
　　　　　**if** (*p_point* > *Max_Distances*)
　　　　　　　**Null** *p_point(x,y,z)* in *PointClouds* data
　　　　　**end if**
　　　　**end for**
　　　　**end if**
　　**if** (*step scheduler* != *Period_ratio_time_on*)
　　　　**for** all *p_point* in *PointsCloud_ℱ* **do**
　　　　　**if** (*p_point* > *Min_Distances*)
　　　　　　　**Null** *p_point(x,y,z)* in *PointsCloud_ℱ* data
　　　　　**end if**
　　　　**end for**
　　　　**end if**
　　　　*Step scheduler = Step scheduler + current_time*
　　　　**return** *PointsCloud_ℱ*
**end function**

### 3.2. Local SLAM and Map Construction

The map generated by Cartographer consists of two parts. The first part is a map representation based on a local coordinate known as submap. The second part is a global map representation constructed from a combination of accumulated submaps which are, then, also corrected or repaired whenever loop closure is identified, as described in Section 3.5. The process of generating submap is an iterative process that compares scans from the LiDAR sensor to a submap coordinate frame. The pose transformation of the scan frame $\mathcal{F}$ in the submap is represented as $T_{\mathcal{F}}$ as defined in Equation (1):

$$T_{\mathcal{F}}p \;=\; \underbrace{\begin{pmatrix} cos\mathcal{F}_\theta & -sin\mathcal{F}_\theta \\ sin\mathcal{F}_\theta & cos\mathcal{F}_\theta \end{pmatrix}p}_{rotation} \;+\; \underbrace{\begin{pmatrix} Fx \\ Fy \end{pmatrix}}_{translation} \tag{1}$$

Submap is created with the representation of a probability grid form. An object is identified in the map after passing a miss and hit algorithm on the scan [1]. Multiple consecutive scans from LiDAR are necessary to build the submap. Then, every hit of the LiDAR scans is inserted on a submap grid point associated with an intersected pixel. Meanwhile, every miss is inserted on the grid associated with the pixel which excludes the points which already have the hit. An illustration of the submap grid is described in Figure 3. The hit probability $p_{hit}$ is updated in Equation (2). It must be noted that $Weight_{map\_size}$ in a large sensor scan must be larger than a smaller scan because it contains more feature information. The similar equation is also used in the probability miss calculation, $p_{miss}$.

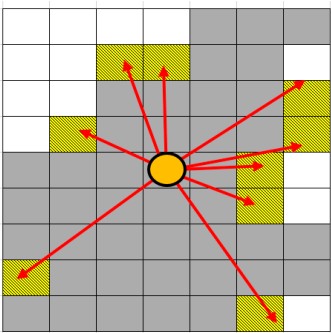

**Figure 3.** Miss and hit probability from a sensor scan to map pixel area.

$$odds(p) = \frac{p}{1-p^{\circ}} * Weight_{map\_size}$$

$$M_{new}(x) = clamp(odds-1(odds(M_{old}(x)).odds(p_{hit}))) \tag{2}$$

### 3.3. Ceres Scan Matching

SLAM uses a scan matcher to process sensor data and to obtain pose. Then, the pose obtained is used as a reference coordinate to insert scan data into the submap. This estimated pose is computed using a comparison between current scan states of LiDAR and submap. During vehicle motion, the scan always has familiarity with the previous scan.

Moreover, a scan matcher filters the residual difference of these consecutive scan data and obtains the translation with the rotation to estimate the pose. The scan matcher responsible in this SLAM is called a Ceres scan matcher [24], which works to find optimal probability values in scan point in the grid-based submap. In the process of scan matching to the submap, due to the presence of multi-distance scan scheduler, it is unnecessary to use a larger search window to match the scan into the submap. The smaller scan distance requires a smaller search window size which increases processing speed.

When the scan uses a maximum LiDAR distance and generates $\mathcal{F}_{max}$,

$$\underset{\mathcal{F}_{max} \in W_{max}}{argmin} \sum_{k=1}^{K} (1 - M_{smooth}(T_{\mathcal{F}} h_k))^2 \tag{3}$$

When the scan uses smaller LiDAR distance and generates $\mathcal{F}_{min}$,

$$\underset{\mathcal{F}_{min} \in W_{min}}{argmin} \sum_{k=1}^{K} (1 - M_{smooth}(T_{\mathcal{F}} h_k))^2 \tag{4}$$

where $h_k$ is a scan frame transformed by $T_{\mathcal{F}}$ to the submap frame, $M_{smooth}$ is a bi-cubic interpolation smooth filter for probability values of submap, $W_{max}$ and $W_{min}$ are the search window areas of the scan matcher where their size follows the size of frame $\mathcal{F}$ which contains the width and height applied around the last transformed frame pose $h_k$ location.

### 3.4. Adaptive Performance Control

The AMDS scheduler, proposed in this paper, combines the multi-distance scheduler approach with some adaptive methods to further reduce sensor scan sizes, scan matcher search window sizes, and iteration for pose and submap generation based on SLAM computational power load. If the load is heavy where the system cannot achieve the real-time online computation, the SLAM tries to reduce these variables to reduce the performance penalty. The load weight is obtained by measuring the time needed to generate a pose each time. If the time needed is more prolonged than a specified threshold, it activates the adaptive control. The threshold values are dependent on the sampling rate of Cartographer. A 20 Hz sampling rate of pose generation requires a processing speed less than 50 ms to achieve real-time performance. These threshold values must be set much lower than the sampling rate so that the delay does not overstep the boundary limit of real-time speed. The reason for this adaptive control is that the real-time performance is very crucial in autonomous vehicles. As the SLAM progresses through the unknown lane, building more maps, the SLAM is burdensome as the map grows [25]. In addition, the accuracy of the map and pose likely have a low impact when using adaptive methods to reduce the scan matcher, as the accurate initial map obtained from the initial SLAM is still utilized to create a bigger map. Similarly, a good initial map computation can result in correct end pose computation. Figure 4 shows the algorithm of the proposed method:

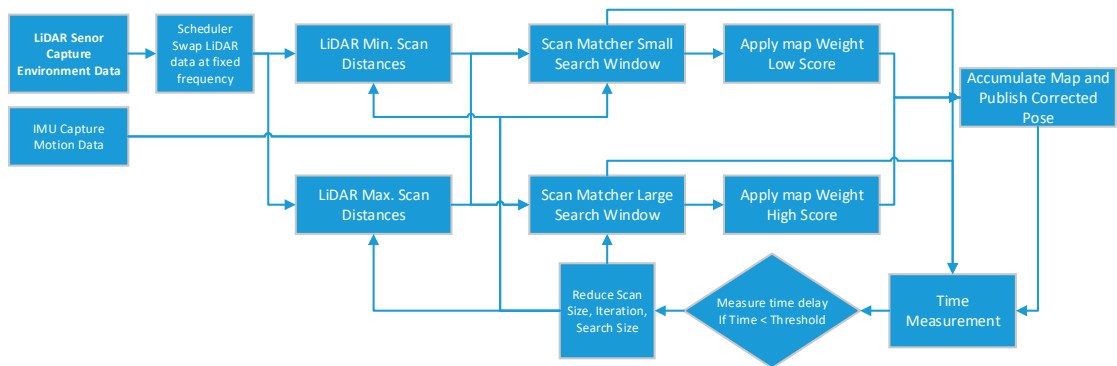

**Figure 4.** Adaptive multi-distance control applied on local SLAM.

### 3.5. Global SLAM Loop Closure

The process to estimate pose in local SLAM always generates errors caused by the presence of grid map resolutions, sensor noises, and different map features which affect the accuracy. Although the error is small, after some prolonged distance of travel, it accumulates and creates a bigger error because the calculation is based on the total sum of every previous translation and rotation. The drift error becomes bigger as the travel distance is accumulated. By comparison, the global SLAM process tries to reduce this error using the optimization method called sparse pose adjustment (SPA) technique [26]. In this optimization method, a similar scan matcher in local SLAM is used but in a more extensive map range to cover the accumulated submaps (global map) for pose correction.

#### 3.5.1. Optimization Problem

The optimization problem is based on loop closure detection, which is a process to recognize previously visited areas by matching sensor scans against the accumulated submap. The optimization problem is formulated as nonlinear squares. Every few seconds, the Ceres scan matcher is utilized to compute (5).

$$\underset{\mathbb{P}^M \mathbb{P}^S}{argmin} \frac{1}{2} \sum_{ij} \rho(E^2(\mathcal{F}_i^m, \mathcal{F}_j^s; \Sigma_{ij}, \mathcal{F}_{ij}))^2 \tag{5}$$

where the scan pose, $^S = \left\{\mathcal{F}_i^m\right\}_{j=1,\dots,n}$, and submap poses, $^M = \left\{\mathcal{F}_i^m\right\}_{i=1,\dots,m}$, in the world are optimized with some constraints. These constraints are in relative pose form $\mathcal{F}_{ij}$, and their covariance matrices, $\sum_{ij}$. For a pair of submap $i$ and $j$, the pose $\mathcal{F}_{ij}$ describes the coordinate of the submap frame where the scan was matched. The residual, $E$, for such a constraint is computed by:

$$E^2\left(\mathcal{F}_i^m, \mathcal{F}_j^s; \sum_{ij}, \mathcal{F}_{ij}\right) = e\left(\mathcal{F}_i^m, \mathcal{F}_j^s; \mathcal{F}_{ij}\right)^T \sum_{ij}^{-1} e\left(\mathcal{F}_i^m, \mathcal{F}_j^s; \mathcal{F}_{ij}\right), \tag{6}$$

$$e\left(\mathcal{F}_i^m, \mathcal{F}_j^s; \mathcal{F}_{ij}\right) = \mathcal{F}_{ij} - \begin{pmatrix} R_{\mathcal{F}_i^m}^{-1}\left(t_{\mathcal{F}_i^m} - t_{\mathcal{F}_j^s}\right) \\ \mathcal{F}_{i;\theta}^m - \mathcal{F}_{j;\theta}^s \end{pmatrix} \tag{7}$$

$\rho$ loss function using Huber loss is utilized to reduce influence of outliers.

#### 3.5.2. Branch and Bound Scan Matching (BBS)

The algorithm to find a matched scan $\mathcal{F}^*$ is described in Equation (8), where $M_{nearest}$ is map extended to the nearest grid point of the corresponding pixel, and $W$ is the search window.

$$\mathcal{F}^* = \underset{\mathcal{F} \in W}{argmax} \sum_{k=1}^{K} M_{nearest}(T_{\mathcal{F}} h_{\mathcal{F}}) \tag{8}$$

The BBS scan matcher algorithm tries to compute matched scan with the best score by iterating the scan under search window inside map. The search window iteration contains finite step $\delta_\theta$, for rotation and, $r$, for translation (9). The detailed process of scan matcher iterations can be seen in Table 2.

$$w_x = \frac{Wx}{r}, w_y = \frac{Wy}{r}, w_\theta = \frac{W_\theta}{\delta_\theta} \tag{9}$$

**Table 2.** Description of the native algorithm of branch and bound scan matcher.

*best_score* ← **Null**
**for** *jx* = -$w_x$ to $w_x$ **do**
**for** *jy* = -$w_y$ to $w_y$ **do**
**for** *jθ* = -$w_\theta$ to $w_\theta$ **do**
*score* ← $\sum_{k=1}^{K} M_{nearest}(T_\mathcal{F} h_\mathcal{F})$
**if** *score* > *best_score* **then**
match ← $\mathcal{F} + (rjx, rjy, rj\theta)$
*best-score* ← *score*
**end if**
**end for**
**end for**
**end for**
**return** *best-score* and *match*

As described in Table 2, each step of translations and rotations within the search window is tested one by one for evaluation. When the scan matcher finds the matched pair with better score, it will replace the previous matched scan pair with lower score. This technique has a downside where a large sized search window required for SLAM global map will result in quadratic growth of iterations.

The Cartographer technique tries to eliminate the iteration needed for BBS scan matcher using a method called depth first search branch and bound scan matching (DFS BBS) which is resolved by finding a good subsets of search window (*W*) possibilities expressed as nodes in tree. The algorithm of DFS BBS can be seen in Table 3. It is done by computing a good upper bound node with multi-resolution grid maps. The DFS BBS work by downscaling existing submap to several resolution factors of one-quarter and one-half map size, etc., which are then, used as scan matcher inputs. Each selection of search window node will depend on previous node result. The scan matching process uses several layers of node, which starts from lower resolution for preliminary low accuracy computation of estimated translation and rotation. Then, the preliminary result with the best score is used to bypass many iterations of possible translation rotation in higher resolution. This process is repeated until the highest layer node on full resolutions map is reached.

**Table 3.** Depth first search branch and bound scan matcher.

*score_thresold* -> *best_score*
Compute and memorize a score for each element in $C_0$.
Initialize a stack *C* with $C_0$ sorted by score, the maximum score at the top.
**while** *C* is not *empty* **do**
    Pop c from the stack C
    **if** score(c) > *best_score* **then**
        **perform** leaf selection (·)
    **end if**
**end if**

Leaf selection (·)
  **if** *C* is leaf node **then**
    $\mathcal{F}_c$-> match.
    score(c) -> *best_score*.
  **else**
    Branch: split c into nodes $C_c$.
    Compute and memorize a score for each element in $C_c$.
    Push $C_c$ onto the stack *C*, sorted by score, the maximum score last.
  **end if**
**return**

## 4. Experimental Results

In this section, we present two different types of experiments to validate the proposed AMDS Cartographer technique. The first test compares the performance accuracy of the proposed system against the standard Cartographer technique using KITTI Vision Benchmark Suite [27] which is an open public dataset developed by Andreas Geiger et al. that measures SLAM trajectory performances against ground truth measured with a real-time kinematic global positioning system (RTK-GPS). Meanwhile, for the second test, the system was integrated in an actual vehicle to test its capability in a more exclusive area where the KITTI dataset could not provide. It also validates the consistency of the AMDS Cartographer technique with a different hardware setup as compared with the Vanilla Cartographer techniques, NDT-SLAM and LOAM.

### 4.1. KITTI Benchmark

In the first validation test, the KITTI dataset used hardware that consisted of Velodyne 64 LiDAR sensors coupled with the IMU. These sensors were mounted in a station wagon which collects data in urban street areas around Karlsruhe, Germany. In our case, the computing specification to test the SLAM used the AMD quad core 3.2 Ghz CPU installed with a robot operating system (ROS) [28]. The data of the LiDAR from the KITTI were also filtered to use a maximum limit of 60 m scan distances from the original 100 m. This reduced sensor noise and distortion found in the rotating LiDAR during high speed motion [29]. The AMDS was set to use long distance scans of 60 m and short distance scans of 25 m, respectively. Then, the methods were compared to the Vanilla Cartographer technique using the two scan states. The grid size of the map in all cartographer configurations was set to use 0.2 m in resolution.

The accuracy performance analysis of the AMDS was analyzed by comparing the local SLAM performance of AMDS to the Cartographers' performance with fixed distances; as well as the AMDS after the global SLAM optimization. A good local SLAM performance resulted in fewer drift errors which guaranteed that global SLAM more easily recognized the previously visited area. When the drift was too large Cartographer failed to perform optimization. Thus, global SLAM optimization in Cartographer requires different tuning parameters to adapt map conditions and sensor configuration (e.g., scan matcher score threshold, search window size, scan matching iteration, etc.) unlike in local SLAM. For the first experiment, the result using the KITTI dataset is shown in Figure 5.

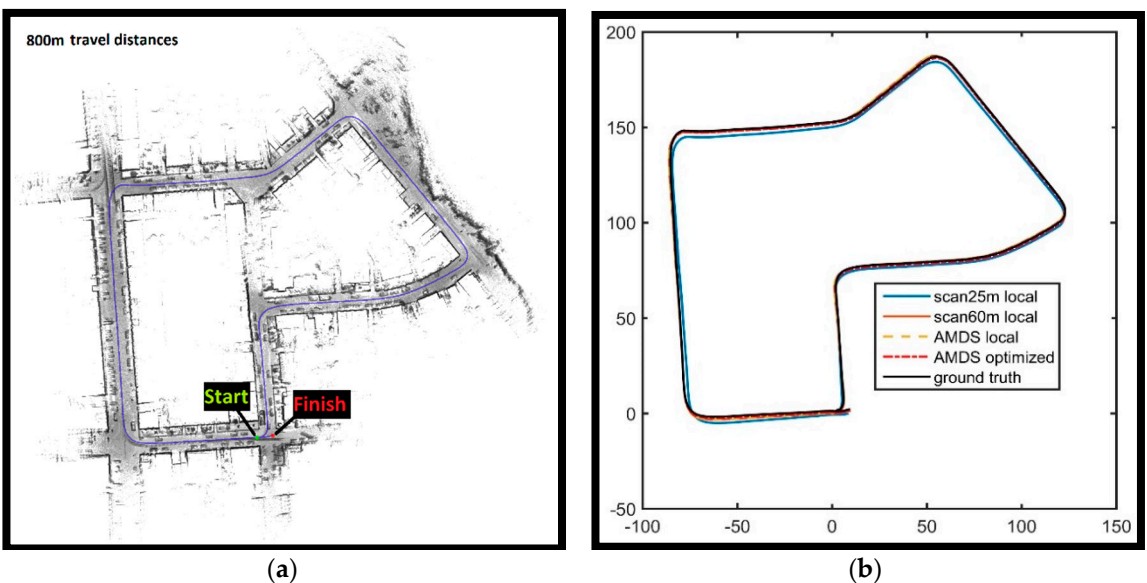

**Figure 5.** Trajectory results of the KITTI benchmark: (**a**) Map location and (**b**) trajectory results.

In the KITTI benchmark, the AMDS (60 m/25 m) local SLAM achieves similar accuracy as compared with Cartographer (60 m scan) local SLAM. The Cartographer (60 m scan) local SLAM achieves maximum error 0.9 m differences against the ground truth while the AMDS (60 m/25 m) have around 1 m, however, the Cartographer (25 m scan) technique produces a higher error around 4.9 m from ground truth. From the analysis, higher errors from Cartographer (25 m scan) are due to the lack of environmental structure sampled. The area used in the KITTI dataset contains rich structure environment, and thus a scan matcher in the Cartographer technique easily matches straight line shapes as the key features in the map. Other features such as vegetation, poles, and tree trunks are also utilized as the Cartographer scan matcher's key frame, but, most of the time, structures such as walls are easily identified by the scan matcher to connect building lines. The results of the scan matcher in smaller scans, however, do not have enough information to match the lines of structures against building lines at far distances. As a result, the scan matcher cannot provide accurate judgments when estimating motion continuously. Unlike the Cartographer (25 m scan) technique, the AMDS (60 m/25 m) suffers slightly from this because the AMDS is still utilizing a large scan to provide the scan matcher with more accurate motion judgments. Meanwhile, the global optimization of the Cartographer technique, when using the AMDS, successfully reduces the impact of drift error from 1 m error from ground truth to 0.5 m. For the mean error against ground truth, AMDS (60 m/25 m) has an average of 0.36 m, whereas AMDS with global optimization reduces the mean error to 0.2 m. The result of error against ground truth can also be seen from the map generated, as shown in Figure 6.

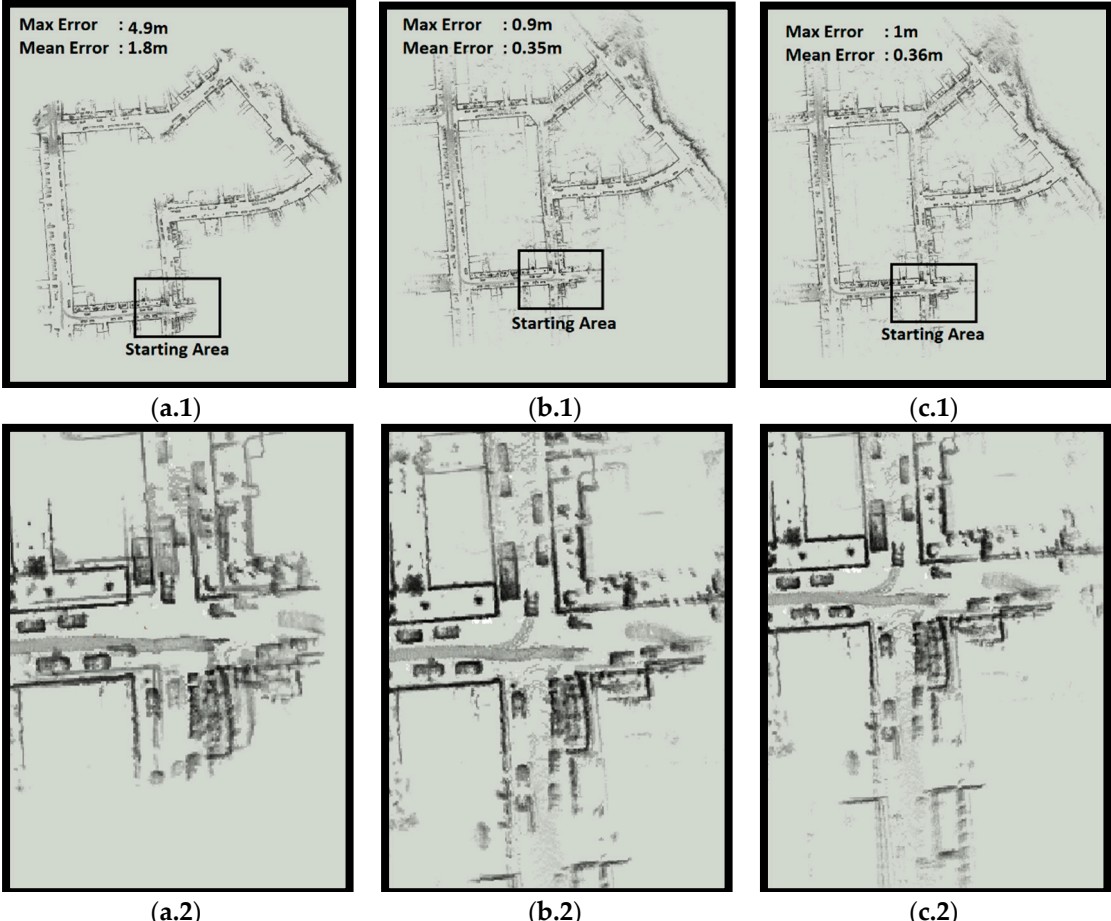

**Figure 6.** KITTI map results: (**a**) 25 m scan, (**b**) 60 m scan, (**c**) AMDS (60 m/25 m), (**1**) full map, and (**2**) zoomed starting point.



In Figure 6, the data obtained from the 25 m scan gave higher errors in positions from the known ground truth. This is seen from the starting point area where the building structures are not properly aligned. The beginning or initial submap data are overlapped with the final submap data, and thus we can see two identical walls overlapping in the starting area. Meanwhile, the AMDS had a difference of one meter from the known ground truth, and the map, as shown in Figure 6c.2, displays that the overlapping structure is almost invisible. A similar result also applies for the 60 m scan distances in Figure 6b.2, where it had a difference of one meter from the known ground truth. Although the AMDS (60 m/25 m) has a similar result as Cartographer (60 m scan), it successfully utilized better time performance in pose generation. In KITTI validation, the local AMDS (60 m/25 m) required 25 ms to generate pose while the local Cartographer (60 m scan) required 27 ms which shows approximately an eight percent reduction in the time performance.

### 4.2. Experimental Validation

The second experiment tested the platform in exclusive road with various conditions. The hardware test-bed setup used Ouster OS1 LiDAR 64 layer together with an inertial measurement unit that modelled Spatial IMU by Advance Navigation. The field test was conducted in an actual environment using an MPV equipped with a LiDAR system. The computing specification used an AMD quad core 3.2 GHz processor. Figure 7 shows the experimental test-bed setup.

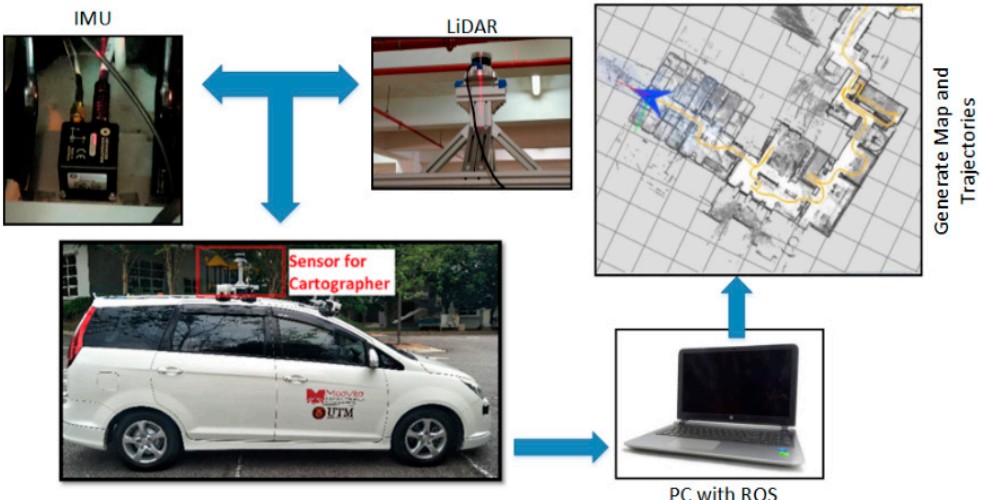

**Figure 7.** An MPV sized vehicle for SLAM configuration.

In the second experiment, since the hardware used a custom setup, the sensor data could be easily adjusted to connect to other SLAM methods for comparisons, for example, NDT-SLAM and LOAM. The experiment was conducted with a similar preset configuration. The performance evaluation, however, did not use the known ground truth because RTK-GPS is very costly. In contrast to the KITTI vision benchmark suite, other researchers have found alternative methods to evaluate SLAM performance. Rainer Kümmerle et al. [30] measured distance differences during the map alignment in the correction and optimization processes of SLAM. The distance difference values in every optimization process were then averaged to find the relationship with SLAM consistency. Good SLAM performances were indicated when the correction and optimization processes do not have high fluctuation or distance differences. However, our work used a simpler approach because the SLAM which does not have optimization, such as LOAM, could not be assessed in such way. The SLAM's accuracy was validated through a closed loop run, where the vehicle must travel the road with the start and finish pose located at the same coordinate. Ideal trajectories generation is where the SLAM can provide identical finish pose at the exact original starting point. Thus, when the trajectories deviated from the actual

position (drift error), they are measured by measuring the distance differences between these two points. The details of the test conducted are shown in Figure 8.

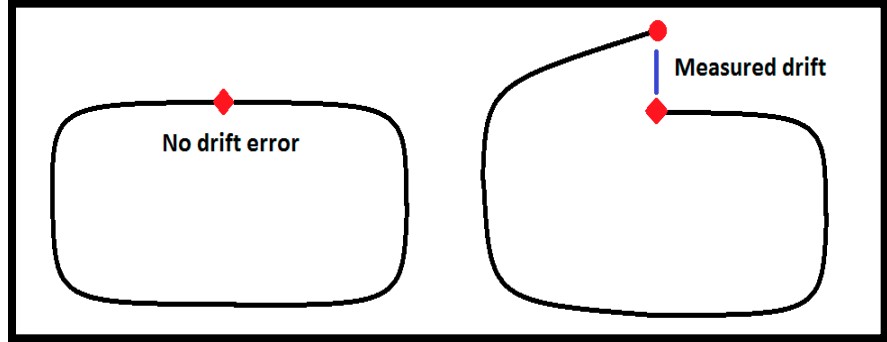

**Figure 8.** Measuring drift error in trajectories.

Test locations for the second experiment were conducted in the urban area of Cyberjaya, Malaysia, as shown in Figure 9.

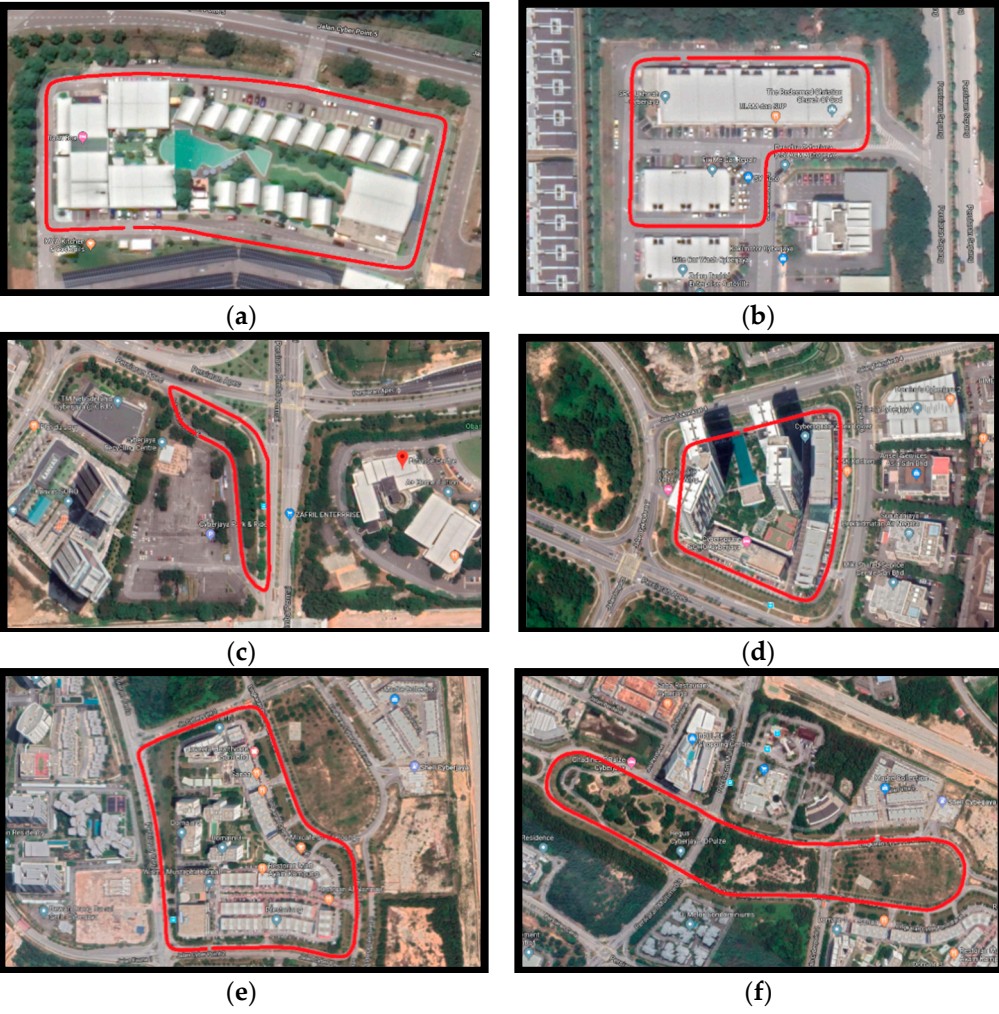

**Figure 9.** Ariel views of test locations: (**a**) Map 1, (**b**) map 2, (**c**) map 3, (**d**) map 4, (**e**) map 5, and (**f**) map 6.

The test was done at a speed limit of 30 km/h to reduce rotating LiDAR sensor distortion at high speed [29]. According to Figure 9, the test locations for the environments of map 1 and map 2 are

located near the residential area, respectively. The road travel distance is around 400 m and 500 m. Map 1 represents a location which has rich features such as building structures, while the second map has many dynamic objects. Map 3 is a road area that represents plenty of vegetation and less feature structures. The travel distance for this area is approximately 480 m. Map 4 is an area with non-flat surfaces. It has an upward and downward slope with a maximum 20 degree road angle. The last two maps, map 5 and map 6, were selected to test the proposed system on wider area locations. The road travel distances are approximately 1.4 km and 2 km, respectively. However, there were some difference purposes for the map selections. Map 5 was selected to test the system in an area with a moderate amount of nearby buildings. Meanwhile, map 6 was selected to test the system in a wide area with fewer structure features nearby. For the accuracy test, the results of the experiment are shown in Figures 10 and 11, which contain data of pose trajectories and map details.

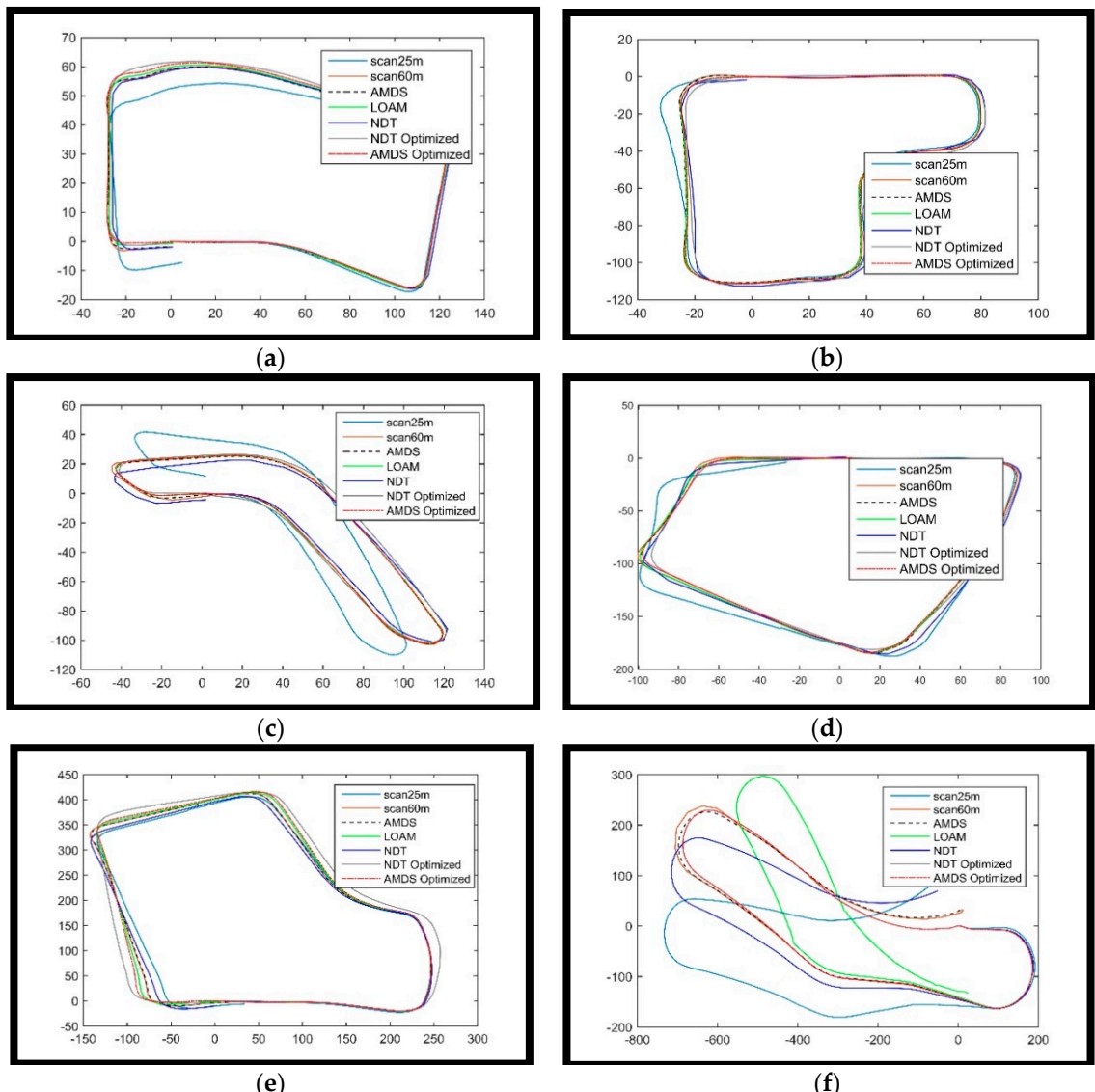

**Figure 10.** Trajectories generation: (**a**) Trajectory results on map 1, (**b**) trajectory results on map 2, (**c**) trajectory results on map 3, (**d**) trajectory results on map 4, (**e**) trajectory results on map 5, and (**f**) trajectory results on map 6.

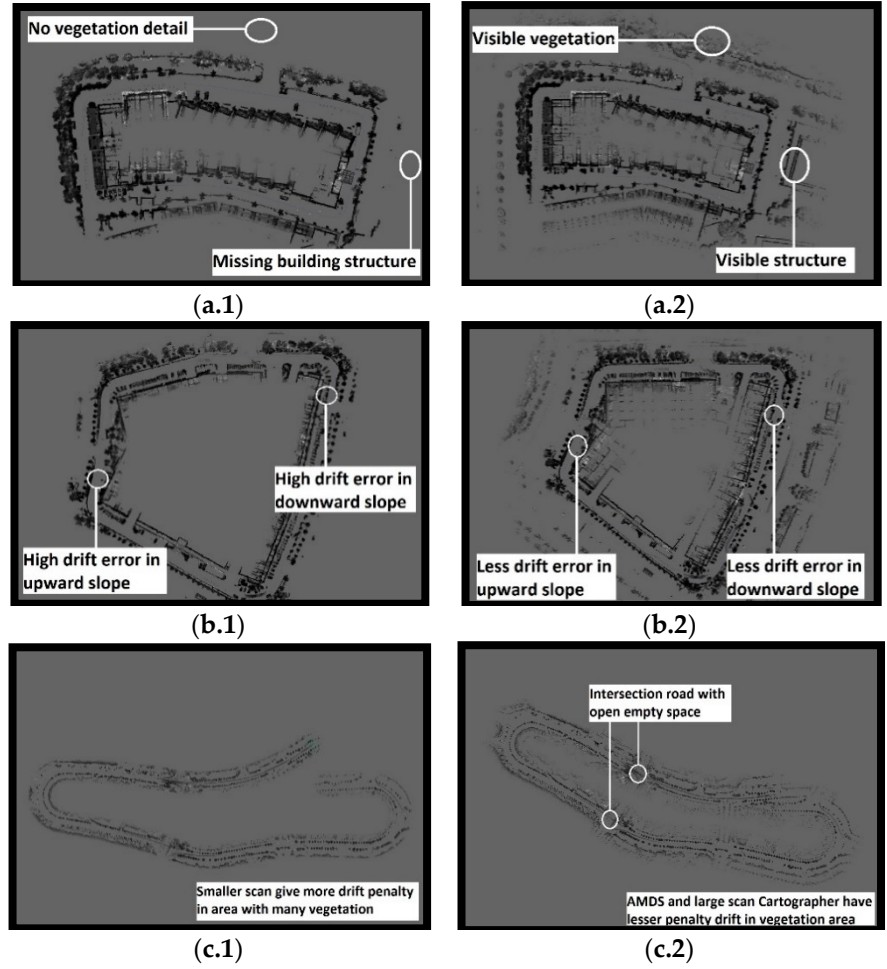

**Figure 11.** Map accuracy and detail: (**a**) Result map 1, (**b**) result map 4, (**c**) result map 6, (**1**) Cartographer using 25 m scan distances, and (**2**) Cartographer using adaptive multistage distance scheduler (AMDS) (60 m/25 m) scan distances.

On the basis of the trajectories generated, as shown in Figure 10, most of the results show the same patterns. The Cartographer local SLAM technique implementation using AMDS (60 m/25 m) results in minimal impact in trajectories as compared with the standard Cartographer (60 m scan) technique. The longest distances traveled in map 6 (2 km distances) result in 2 m distance differences versus Cartographer (60 m scan). The AMDS trajectories after optimization also give less drift error as compared with the heavy computational load with optimization NDT-SLAM, since NDT-SLAM does not perform online mapping. Meanwhile, the LOAM performance falls off in long distance travel due to the unavailability of optimization methods unlike the NDT-SLAM or Cartographer. LOAM also gives poorer results when there is no solid structure on the road nearby, as shown in map 6. The detailed table comparison of drift error can be seen in Table 4.

**Table 4.** Measured drift error translation (m) and rotation (°).

|  | Map 1 | | Map 2 | | Map 3 | | Map 4 | | Map 5 | | Map 6 | |
|---|---|---|---|---|---|---|---|---|---|---|---|---|
|  | m | ° | m | ° | m | ° | m | ° | m | ° | m | ° |
| Carto25 m local/unoptimized | 8.9 | 10.3 | 9.7 | 9.5 | 11.1 | 9.9 | 9.7 | 17.5 | 34.3 | 7.7 | 110 | 35.9 |
| Carto60 m local/unoptimized | 2.0 | 1.4 | 0.4 | 1.3 | 7.6 | 2.6 | 0.5 | 15 | 13.9 | 5.1 | 35 | 9.5 |
| AMDS (60 m/25 m) Local/unoptimized | 2.1 | 1.4 | 0.4 | 1.3 | 7.6 | 2.6 | 0.6 | 1.5 | 15.9 | 5.0 | 37 | 9.6 |

**Table 4.** *Cont.*

| | Map 1 | | Map 2 | | Map 3 | | Map 4 | | Map 5 | | Map 6 | |
|---|---|---|---|---|---|---|---|---|---|---|---|---|
| | m | ° | m | ° | m | ° | m | ° | m | ° | m | ° |
| LOAM Unoptimized | 10 | 0.5 | 1.1 | 0.5 | 2.6 | 1.4 | 1.0 | 0.5 | 5.8 | 5.0 | 135 | 19.9 |
| NDT unoptimized | 1.9 | 1.3 | 2.5 | 2.9 | 4.6 | 2.1 | 3.7 | 1 | 9.6 | 3.7 | 81 | 37 |
| NDT optimized | 0.3 | 0.5 | 0.3 | 0.5 | 0.2 | 0.7 | 0.4 | 0.7 | 0.4 | 2.0 | 0.1 | 1.9 |
| AMDS (60 m/25 m) Optimized | 0.2 | 0.5 | 0.2 | 0.5 | 0.3 | 0.7 | 0.2 | 0.8 | 0.5 | 1.7 | 0.2 | 1.9 |

The analysis of the map generated from AMDS provides detailed information of AMDS accuracy. The overall result of the Cartographer map generation from the experiment highlights the important areas only. The information is shown in Figure 11.

From the map generation, as shown in Figure 11.a, it can be observed that the AMDS (60 m/25 m) still preserves map detail and better accuracy than the 25 m scan. A smaller scan cannot sample feature that is out of scan range. Figure 11a.2 shows the area where structures are missing. In AMDS, the missing gap of a smaller scan are filled by larger scans with the help of scheduler. Thus, the missing building still can be reconstructed inside the map. A smaller scan also has an increased error penalty when the road has an extreme downward and upward slope. The scan matcher judgment of Cartographer performs poorly as it does not have enough feature information to increase pose confidence when aligning the scan in a 2D map. Performance of the AMDS (60 m/25 m) has better results in this area, as shown in Figure 11b.2. In addition, map with upward downward roads and vegetation also contributes to the increased error of Cartographer. In long distance travel, the 25 m scan performs even more poorly. Similar bad drift results also apply to NDT-SLAM that does not use global optimization in vegetation areas. Map 6 highlights a 2 km total distance traveled and the result for the 25 m scan Cartographer drift of approximately 110 m from the starting position point. The result is visible in Figure 11c.1, whereas Figure 11c.2 shows reduced drift errors. It is no0ted that the LOAM has a massive performance drop in map 6, as shown in Table 4 and Figure 10f. The LOAM started to drift heavily at areas around road intersections and these areas have large empty open spaces. This kind of area is not suitable for LOAM to extract specific feature information such as edge lines and planar surfaces and eliminates other information.

To show the comparison of complete pose generation time of AMDS, NDT, LOAM, we tested Cartographer in various urban road environments. The average results are shown in Figure 12. The proposed AMDS in Cartographer boosts speed up to 16% faster than the standard 60 m scan Cartographer. The processing speed also varies based on travel distance. AMDS with 0.5 km travel distances generates pose faster, at approximately 21 ms. This increases linearly as the maps become larger. As shown in Figure 12, 1 km, 1.5 km, and 2 km travel distances generate poses at approximately 23 ms, 27 ms, and 34 ms respectively. The AMDS Cartographer technique also has several advantages as compared with other SLAM methods such as NDT and LOAM. On the one hand, in terms of speed performance, it is several times faster than NDT-SLAM as it can generate map data online. Map generation in the NDT-SLAM, on the other hand, needs to be processed offline before it can be plotted. Meanwhile, the accuracy of Cartographer in long distance travel is better than the LOAM, as LOAM does not have a map optimization (loop closure) algorithm, as previously shown in Figure 10.

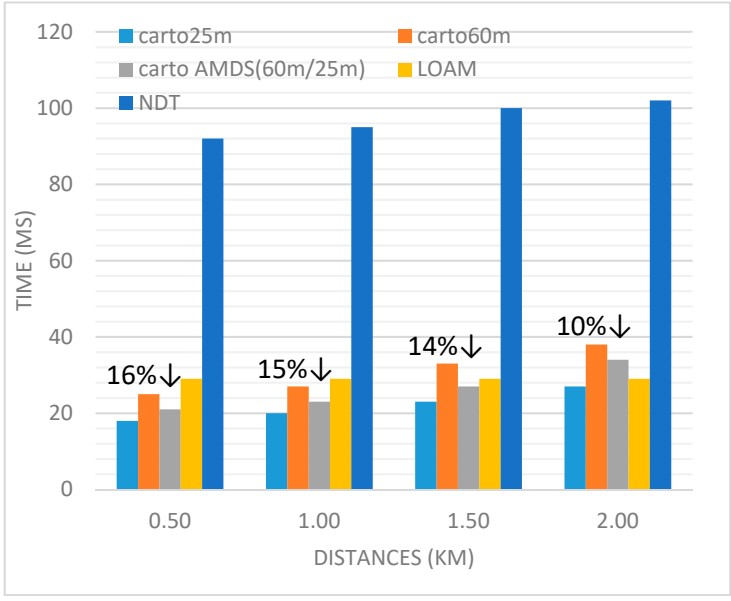

**Figure 12.** Average speed performance comparison.

## 5. Conclusions

Cartographer is a grid map-based SLAM which uses Ceres-based scan matchers for pose generation. It is known as a SLAM with wide sensor support and flexible configuration as compared with other SLAM such as NDT or LOAM. The multistage distance scan scheduler is a program that limits sensor ranges to reduce the computational burden of scan matchers. By swapping the scan distance of the sensor between a smaller range (25 m) and a longer range (60 m) and adaptively limiting the search size of the scan matcher to handle different scan sizes, it improves pose generation performance time up to 16% as compared with a fixed scan distance of 60 m. Meanwhile, it is also able to produce similar accuracy as compared with a fixed scan distance of 60 m in terms of pose generation. The map generated size is also similar as compared with a 60 m scan distance. Thus, its benefits include improved overall performance while maintaining accuracy.

**Author Contributions:** Conceptualization, A.D.; methodology, A.D., M.A.A.R., and H.Z.; formal analysis M.A.A.R., M.H.M.A., and W.M.H.W.A.; data curation A.D., W.M.H.W.A.; writing-original draft A.D.; writing-review and editing M.A.A.R., M.H.M.A.; visualization A.D., W.M.H.W.A.; supervision M.A.A.R., M.H.M.A., and H.Z. All authors have read and agreed to the published version of the manuscript.

**Funding:** This research was supported by Vehicle system lab, Malaysia Japan Institute of Technology (MJIIT) specifically by U.T.M i-Drive team. This work is funded by Ministry of Higher Education Research University Grant, under fundamental research grant scheme (FRGS), cost center no. 5F204.

**Acknowledgments:** The author also would like to acknowledge Moovita, Pte Ltd for their knowledge sharing and suggestions to improve researches quality.

**Conflicts of Interest:** The authors declare no conflict of interest.

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
