# Peer review of "Cartographer SLAM Method for Optimization with an Adaptive Multi-Distance Scan Scheduler"

_applsci, doi:10.3390/app10010347_

Round 1

Reviewer 1 Report

Writing

In general, the writing should be checked. There are several phrases that are really difficult to understand and many grammatical errors.

Review

Drift error does not only occur due to “sensor accuracy specification”. Sensor measurements may change because of electrical noise and other causes, temperature, etc. This fluctuation can be either systematic or random.

Perhaps it would be more accurate to use to “filter sensor distances” rather than “control sensor distances” because the authors do not change any sensor parameter to modify the sensor outputs or its performance. Once the sensor deliver the 2D or 3D points, the authors filters the data to decide which ones will be used.

Regarding the method data diagram, the authors evaluate the time consumption of the scan matcher to estimate the vehicle pose. If that time exceeds a time_threshold the adaptive process is activated to reduce that processing time and achieve the desire frequency. Have the authors detected or identified a minimum value (time/distance) in which the pose accuracy is of out range?

The Global optimization and scan matching algorithms are the implementation provided by the Google Cartographer method. The fore, the authors approach is only related to the adaptive multi distance control which is applied in the local SLAM stage. So, the key issue of this work should be to present a relevant experimental results and discussion. The idea to compare AMDS Cartographer with the Google Cartographer and LOAM NDT_SLAM is good. However, it should provide more in-depth analysis.

The implementation in ROS is very straightforward, therefore the sensor fusion and data synchronization is solved. The main objective should be focused on details, % of the drift reduction with AMD, how many msec or sec are reduced with the proposal? Is the drift error higher in lateral or longitudinal? This is a very important problem for autonomous vehicles, lateral errors can easily lead to collisions while performing evasive maneuvers or curves, or when passing near a parked vehicle. Longitudinal errors can be compensated with collision sensors. For instance:

The map accuracy evaluation:

which visual analysis is done? What do you mean with “where the maps detail can be seen”, Which is the ground truth map to be compared with? I think the map accuracy should be only be tested with visual assessment.

Accuracy of the trajectory → drift error:

Why do the authors selected 25 and 60 meter to evaluate the Cartographer algorithm, Why was the maximum distance acquired by the sensor not used? Or is 60m the maximum distance? If not, then your method will have indirect advantage over the standard method, because the adaptive control uses the maximum distance to estimate the proper translation or rotation in a given situation better than a limited distance. What is the Lidar you are using?

If the authors are going to compute the Global Map optimization of the AMD approach and do not compared it with other similar results, I do not think it should be added into Table 5. The authors should compute the Global Map optimization of the standard solution and then, to compare both. According to the introduction section, this paper is intended to show the implementation of the Google Cartographer, so this is not out of the focus of this work. Otherwise, I do not thinks these results (global optimization) show the improvements of the proposal. It does not clarify whether the good results are due to the AMD approach or Google method itself.

Please check the plots legends in Figure 8, what is data1?

Reviewer 2 Report

Please see the attachment for detail.

Round 2

Reviewer 1 Report

The authors have included the KITTI data set to evaluate their method and have incorporated a more in-depth and meaningful analysis of their results. The modifications in table 5 show the results more clearly. Although the authors' proposal does not show a considerable improvement with respect to the optimized NTD, in terms of accuracy, it does reduce its computation time.

The efforts made in the experimentation section and the improvements in writing, table and images, make the article interesting for readers.